# Involvement in Fertilization and Expression of Gamete Ubiquitin-Activating Enzymes UBA1 and UBA6 in the Ascidian *Halocynthia roretzi*

**DOI:** 10.3390/ijms241310662

**Published:** 2023-06-26

**Authors:** Hitoshi Sawada, Shukumi Inoue, Takako Saito, Kei Otsuka, Maki Shirae-Kurabayashi

**Affiliations:** 1Sugashima Marine Biological Laboratory, Graduate School of Science, Nagoya University, 429-63 Sugashima, Toba 517-0004, Japan; 2Department of Food and Nutritional Environment, College of Human Life and Environment, Kinjo Gakuin University, Omori 2-1723, Moriyama-ku, Nagoya 463-8521, Japan; 3Department of Applied Life Sciences, Faculty of Agriculture, Shizuoka University, Shizuoka 422-8529, Japan; 4Shizuoka Institute for the Study of Marine Biology and Chemistry, Shizuoka University, Shizuoka 422-8529, Japan; 5Department of Life Science, Faculty of Life Science, Gakushuin University, 1-5-1 Mejiro, Toshima-ku, Tokyo 171-8588, Japan

**Keywords:** ubiquitin, FAT10, ubiquitin-activating enzyme, UBA, E1, fertilization, ascidian

## Abstract

The extracellular ubiquitin–proteasome system is involved in sperm binding to and/or penetration of the vitelline coat (VC), a proteinaceous egg coat, during fertilization of the ascidian (Urochordata) *Halocynthia roretzi*. It is also known that the sperm receptor on the VC, HrVC70, is ubiquitinated and degraded by the sperm proteasome during the sperm penetration of the VC and that a 700-kDa ubiquitin-conjugating enzyme complex is released upon sperm activation on the VC, which is designated the “sperm reaction”. However, the de novo function of ubiquitin-activating enzyme (UBA/E1) during fertilization is poorly understood. Here, we show that PYR-41, a UBA inhibitor, strongly inhibited the fertilization of *H. roretzi*. cDNA cloning of *UBA1* and *UBA6* from *H. roretzi* gonads was carried out, and their 3D protein structures were predicted to be very similar to those of human UBA1 and UBA6, respectively, based on AlphaFold2. These two genes were transcribed in the ovary and testis and other organs, among which the expression of both was highest in the ovary. Immunocytochemistry showed that these enzymes are localized on the sperm head around a mitochondrial region and the follicle cells surrounding the VC. These results led us to propose that HrUBA1, HrUBA6, or both in the sperm head mitochondrial region and follicle cells may be involved in the ubiquitination of HrVC70, which is responsible for the fertilization of *H. roretzi*.

## 1. Introduction

Intracellular short-lived or abnormal proteins are degraded by the ubiquitin–proteasome system (UPS) [1,2,3,4,5,6]. In this system, target proteins are tagged with ubiquitin (Ub), an 8.5-kDa highly conserved protein, by the sequential action of ubiquitin-activating enzyme E1, ubiquitin-conjugating enzyme E2, and ubiquitin ligase E3 in an ATP-dependent manner [1]. Ubiquitinated proteins are degraded by the 26S proteasome in an ATP-dependent fashion [3]. In the first step, the active site Cys residue of E1 forms a thioester bond with the C-terminal Gly residue of Ub using the energy of ATP hydrolysis. This thioester bond is transferred to an active site Cys residue of E2. Then, Lys residues of target proteins are ubiquitinated, forming an isopeptide bond between the carboxyl group of the C-terminal Gly residue of Ub and the ε-amino group of Lys residues of substrate proteins. The final step is catalyzed by E3 (HECT-type E3) or the E2-E3 (RING-type) complex [5,6]. It was recently found that Ser, Thr, and Cys residues and α amino groups of substrate proteins are also ubiquitinated [7,8].

In addition to Ub, several ubiquitin-like proteins (UbLs) have been reported (see reviews, [7,8]). UbLs exhibit sequence similarity to Ub, having a Ub/UbL β-grasp fold structure. Ub/UbL are classified into two groups: one is a Ub-domain-containing protein, and the other is a Ub-like modifier (ULM) [8,9]. While UbLs contain Ub-like domains involved in protein–protein interactions, ULMs are covalently bound to a target protein via isopeptide bonds. Most ULMs, except human leukocyte antigen (HLA)-F-adjacent transcript 10 (FAT10), are involved in intracellular translocation, enzymatic turnover, signal transduction, or transcriptional activity [5,7,8,9,10,11,12,13,14]. Exceptionally, FAT10 (also known as ubiquitin D), a diubiquitin-like protein with C-terminal Gly–Gly residues, is a primary signal for protein degradation, similar to Ub [10,13]. Whereas Ubiquitin Activating Enzyme (UBA)1/E1 can solely activate Ub, UBA6/E1L2 is capable of activating both Ub and FAT10 in an ATP-dependent manner, forming a thioester bond. Both UBA6~Ub and UBA6~FAT10 can transfer Ub or FAT10 to UBA6-specific E2 (USE1) [15], and then Ub or FAT10 are covalently linked to substrate proteins. Both ubiquitinated and FAT10ylated proteins are degraded by the 26S proteasome. A marked difference is that the conjugated FAT10 itself is also degraded by the 26S proteasome, whereas Ub is deubiquitinated and recycled [7,8,9,10,11,12,13,14,15]. This indicates that the 26S proteasome can recognize not only Ub but also FAT10 as a degradation signal. E1, E2, and E3 enzymes for Ub or ULM are known to be structurally and enzymatically related to each other.

It has been reported that the *UBA6* gene is located in the MHC region, and its expression is restricted to vertebrates [15]. However, *UBA6* or *UBA6*-like genes are predicted in the genome database in invertebrates [11], and transcription in sea urchins has been reported (https://www.ncbi.nlm.nih.gov/gene/575276 (accessed on 15 May 2023)). UBA6 is reported to be essential for embryonic development, the immune system, and neuronal functions in mice [13,15]. However, the necessity of UBA6 is still unclear except in mammals, since FAT10 is reported to be expressed only in mammals and is induced by INFγ and TNF-α [13,15]. In addition, it is notable that little is known about the extracellular Ub/ULM-conjugating enzyme system, whereas the Ub/ULM systems play important roles within cells.

Sawada and his colleagues have been studying the sperm proteases involved in fertilization using the solitary ascidian *Halocynthia roretzi* since this aqua-cultured animal is commercially available and gametes can be easily obtained. Generally, ascidians (Urochordata) are hermaphroditic animals that release sperm and eggs nearly simultaneously to the surrounding seawater during the spawning season. However, at least several species, including *Halocythia roretzi* and *Ciona intestinalis* type A (another name; *Ciona robusta*), acquire a self/nonself gamete recognition system to avoid self-fertilization. *H. roretzi* shows strict self-sterility to intact eggs but not to vitelline-coat (VC)-free eggs. Therefore, after sperm attach and recognize the VC of the egg as a nonself-egg, the sperm lytic agent lysin of the VC appears to be activated and makes a small hole, through which sperm penetrate for gamete fusion.

It has been reported that the sperm proteasome and two sperm trypsin-like proteases, acrosin and spermosin, are involved in fertilization, particularly in sperm binding to and/or penetration of the VC [16,17,18,19,20,21,22,23]. In addition, it was also revealed that a 70-kDa sperm receptor protein (HrVC70) on the VC is ubiquitinated by a 700-kDa extracellular ubiquitin-conjugating enzyme complex (exUBC) released upon sperm reaction [21,22,24], an event of sperm activation on the VC characterized by vigorous sperm movement, sperm mitochondrial swelling, translocation through flagellum and eventual shedding [25]. Sawada et al. proposed that sperm extracellular UPS must function as a VC lysin in *H. roretzi* for the following reasons. First, the proteasome is activated and released upon sperm reaction [20,22]. Second, anti-proteasome antibodies and anti-multi-Ub monoclonal antibodies, which show no cross-reactivity to free Ub, and several proteasome inhibitors can inhibit fertilization [21,22]. Third, HrVC70, a self/nonself-recognizable sperm receptor [26] and a target for the sperm proteasome [21,22], is multiubiquitinated by sperm, as revealed by Western blotting. Fourth, the VC of fertilized eggs was immunostained by fluorescently labeled anti-multi-Ub antibodies [21]. Fifth, the exUBC released upon sperm reaction can ubiquitinate HrVC70 but not lysozyme, a widely used substrate for ubiquitination [24]. It was also revealed that Lys234 of HrVC70 is a target residue to be ubiquitinated by site-directed mutagenesis experiments [21]. To function as a lysin, sperm UPS should be localized on the sperm surface as an active form at the necessary time upon sperm activation on the VC. However, the molecular entity of the extracellular ubiquitinating enzyme system during ascidian fertilization is poorly understood.

To address whether E1 activity plays a key role in the process of ascidian fertilization, the effect of an E1 inhibitor on fertilization was first investigated. Since the participation of E1 in the fertilization of *H. roretzi* was suggested, cDNAs of E1 enzymes expressed in the testis were cloned, and the domain architecture and 3D structures were predicted. Localization of E1s was also investigated by immunocytochemistry using affinity-purified antibodies.

## 2. Results

### 2.1. Effect of E1 Inhibitor on Fertilization

To determine whether ubiquitin-activating enzyme E1 plays a role during the fertilization process, the effects of PYR-41, an E1 inhibitor, on the fertilization of *H. roretzi* were investigated according to the procedure described previously [16,27]. As shown in Figure 1, PYR-41 strongly inhibited fertilization in a concentration-dependent manner: PYR-41 showed significant inhibition at 10 µM and strong inhibition at 20 µM (Figure 1). In contrast, the vehicle (0.03% and 0.06% DMSO, respectively) showed no effect on fertilization. These results indicate that ubiquitin-activating enzyme E1 plays a key role in the fertilization of the ascidian *H. roretzi*. These results support our previous observation that the UPS plays a key role in *H. roretzi* fertilization [20,21,22,23,24].

### 2.2. Identification of the UBA1 and UBA6 Genes in H. roretzi

Since the participation of UBA/E1 in *H. roretzi* fertilization was suggested, isolation of a cDNA clone of this enzyme was carried out. First, a gene model of UBA/E1 was explored using the ascidian genome database ANISEED (https://www.aniseed.cnrs.fr (accessed on 15 February 2015). By this analysis, the following seven candidate gene models of UBA were nominated (see Table 1). Possible genes annotated by ANISEED are also listed in Table 1.

Whereas UBA1 is a canonical E1 enzyme of Ub, the E1s of ULM are as follows: UBA2 is involved in the conjugation of a small ubiquitin-like modifier (SUMO) by forming an E1 complex with SAE1 (SUMO-activating enzyme 1), UBA3 is involved in conjugation of neuronal-precursor-cell-expressed developmentally downregulated protein-8 (NEDD8) with NEDD8-activating enzyme 1 (NAE1), UBA4 is involved in conjugation of ubiquitin-related modifier 1 (URM1), UBA5 is involved in conjugation of ubiquitin-fold modifier 1 (UFM1), UBA6 is involved in FAT10ylation and ubiquitination, and UBA7/UBE1L is involved in conjugation of the interferon-stimulated gene, 15 kDa (ISG15). Among these Ub/ULM-activating enzymes, UBA6 exceptionally catalyzes protein modification with Ub and FAT10 [10,12,14]. Since Ub-activating enzyme(s) appears to be involved in ascidian fertilization, UBA1, and UBA6 were focused on.

Although the abovementioned seven gene models showed some homology to UBA1, the molecular phylogenetic tree of *UBAs* revealed that the above #6 and #1 gene models were candidate *UBA1* and *UBA6*, respectively. To clarify the nucleotide sequences of the genes of *HrUBA1* and *HrUBA6*, cDNAs of *HrUBA1* and *HrUBA6* were cloned from the testis according to the procedures described in Section 4 using the primers indicated in Table 2. The evolutionary relationships between UBAs are shown in the molecular phylogenetic tree, which was constructed from the amino acid sequences of ascidian, human, and mouse UBA1 and UBA6, together with human and mouse E1s for ULMs (UBA2, UBA3, UBA4, UBA5, and UBA7) (Figure 2). HrUBA1 and HrUBA6 were classified into the UBA1 and UBA6 clades, respectively.

The amino acid sequences of HrUBA1 and HrUBA6 deduced from the nucleotide sequences of the cloned cDNAs were aligned with human UBA1 and UBA6, respectively, and are depicted in Appendix A. *H. roretzi UBA1* and *UBA6* showed a single open reading frame: HrUBA1 and HrUBA6 consist of 1085 and 1061 amino acids. Furthermore, the estimated molecular mass of HrUBA1 is 122 kDa, and that of HrUBA6 is 119 kDa. The identity between these two proteins was 34.6%.

As indicated in Appendix A, the primary and secondary structures of ascidian HrUBA1 and HrUBA6 were very similar to those of human HsUBA1 and HsUBA6, respectively. From these data, respective domain architectures and predicted 3D structures by AlphaFold2 of HrUBA1 and HrUBA6 are depicted in Figure 3 based on the structures of human UBA1 and 6 [28,29]. HrUBA1/6 are composed of an N-terminal inactive and active adenylation domain (IAD), the first catalytic cysteine half domain (FCCH), the 3-helix bundle (3HB), the active adenylation domain (AAD), the second catalytic cysteine half domain (SCCH) and the ubiquitin fold domain (UFD), in common with human HsUBA1/6. The models of HsUBA1 and HrUBA6 were compared with the crystal structures of HsUBA1 (PDB ID: 6DC6) and HsUBA6 (PDB ID: 7PVN), as shown in Appendix A [28,29].

### 2.3. Transcriptional Levels of HrUBA1 and HrUBA6 in Several Organs

The *HrUBA1* and *HrUBA6* genes were amplified by using specific primers and template cDNA obtained from various organs or tissues, including the testis, ovary, gill, intestine, and muscles, in *H. roretzi*. To rule out the possible contamination of genomic DNA, short *EF-1α* gene fragments were amplified by PCR using the respective cDNA preparation as a template. The results showed that the cDNA of *EF-1α* was amplified at the expected size (363 bp), suggesting that there was no appreciable contamination of genomic DNA (Appendix A). Then, quantitative PCR (real-time-PCR) analyses for *HrUBA1* and *HrUBA6* were carried out in several tissues/organs using specific primers for these cDNAs. *EF-1* gene expression was used as a reference. The amount of real-time PCR products was expressed as a relative value, defining the amount in the testis as 1.0. As shown in Figure 4, transcription of both the *HrUBA1* and *HrUBA6* genes was detected in the testis. In addition, it was found that both genes were transcribed not only in the testis but also in other organs/tissues. In particular, both mRNA levels in the ovary were prominently higher than those in other organs, including the testis.

### 2.4. Localization of HrUBA1 and HrUBA6 in Sperm and Eggs

To investigate the localization of HrUBA1 and HrUBA6, respective peptide antibodies against HrUBA1 and HrUBA6 were prepared by immunizing rabbits with peptide-linked keyhole limpet hemocyanin. The antigenic peptide regions are indicated in Appendix A, which were selected based on the uniqueness of their sequences and estimated high antigenicity. Respective antibodies showing high reactivities (Appendix A) were purified with the antigenic peptide-immobilized beads and used for immunocytochemistry for HrUBA1 and HrUBA6 (Figure 5 and Figure 6). First, the localization in egg preparation (Figure 5) was examined. As shown in Figure 5A, the anti-HrUBA1 antibody moderately reacted to the VC or follicle cells located over the VC (Figure 5A, upper panels). In contrast, little or no fluorescence was observed in this region by control rabbit IgG (Figure 5A, lower panels). Anti-HrUBA6 antibody weakly reacted to the VC or follicle cells over the VC (Figure 5B upper panels), whereas little or no fluorescence was observed by control rabbit IgG (Figure 5B lower panels). These results suggest that HrUBA1 and HrUBA6 are localized on VC and/or follicle cells.

Next, the localization of HrUBA1 and HrUBA6 on fixed and permeabilized sperm was investigated. Although the reactivity of the anti-HrUBA6 antibody appears to be weaker than that of anti-HrUBA1, the sperm mitochondrial region was stained by both antibodies but not by the control antibody (Figure 6A,B).

## 3. Discussion

In the present report, it was demonstrated that the mRNAs of two ubiquitin-activating enzymes, *HrUBA1* and *HrUBA6*, are expressed in the ovary and testis in the ascidian *Halocynthia roretzi.* Immunocytochemical studies demonstrated that HrUBA1/6 appears to be localized around a sperm mitochondrion, which localizes on the lateral side of the sperm head. It has been reported that the ascidian sperm surface over a mitochondrion binds to the surface of the VC, which allows sperm penetration through the VC using the motive force of mitochondrial sliding through the flagellum [31]. Therefore, localization of the ubiquitin–proteasome system in the mitochondrial region, which is involved in making a small hole for sperm passage of the VC, seems reasonable.

It is currently believed that FAT10, whose gene resides in the MHC region in mammals, is involved in adaptive immunity. Therefore, it is not surprising that FAT10 mRNA has not been identified in invertebrates, including ascidians, because invertebrates lack an adaptive immune system. Although the presence and expression of the *UBA6* gene were clarified by this study, the *FAT10* gene was not identified in the genome database of *H. roretzi*, which encodes a di-ubiquitin structure, lacking a ubiquitin hydrolase-sensitive site such as the RGG-X sequence between two Ub domains.

In mammals, it has already been reported that the E1 inhibitor PYR-41 inhibits fertilization [32]. Our present results coincided well with these results. Since PYR-41 is a cell-permeable inhibitor, the possibility that PYR-41 could inhibit intracellular E1 cannot be ruled out. However, in contrast to the inhibition in cell culture experiments, gametes are exposed to the inhibitor only for several minutes due to the short fertilization time. Therefore, it is thought that only a small amount of inhibitor could penetrate the gamete cell membrane. It is reasonable to consider that gamete surface E1s, HrUBA1 and HrUBA6, may play a key extracellular role in fertilization.

It has been reported that ubiquitin-conjugating enzymes make a 700 kDa complex, which is released from sperm upon sperm reaction induced by alkaline seawater [24]. Because of the limitation of the amount of purified protein, the sequences of its subunits are still unknown. However, the enzymatic properties are considerably different from those of intracellular enzymes, since 10 mM Ca^2+^, a concentration in seawater, is necessary for ubiquitination of HrVC70, and since the enzyme exhibited activity under high salt conditions such as seawater [24]. Furthermore, the enzyme can ubiquitinate HrVC70 but not lysozyme, a widely used substrate for ubiquitination [24]. It is inferred that this complex contains ubiquitin-activating enzyme E1, ubiquitin-conjugating enzyme E2, and ubiquitin ligase E3 since the purified 700 kDa complex catalyzes the ubiquitination of HrVC70 by adding [^125^I]-ubiquitin and ATP-Mg. HrUBA1/6 may be included in this 700 kDa complex. ATP, ubiquitin, ubiquitin-conjugating enzyme complexes, and the proteasome seem to be released from sperm upon sperm reaction since the supernatant fraction after treatment of sperm with alkaline seawater contained ATP and the enzymes involved in ubiquitination and proteolysis [21,24]. After the sperm reaction, clear HrUBA1/6 localization in sperm mitochondria was not detected, but some were localized to the released mitochondria. Although the mechanisms of transportation of the ubiquitin-conjugating complex and the proteasome to the sperm and egg surface are unknown, both sperm and follicle-cell E1 (HrUBA1/6) may be involved in ubiquitination of HrVC70, a sperm receptor, which allows sperm penetration through the VC (Figure 7). Further studies are necessary to elucidate the detailed localization and extracellular functions of the ubiquitin-conjugating enzyme complex, including E1, E2, and E3, during fertilization of the ascidian *H. roretzi.* Although genome editing experiments of E1, E2, and E3 would be difficult due to the inability of culturing a juvenile into a sexually matured adult in an aquarium, several specific inhibitors against the UPS [33], including deubiquitinating enzyme [34,35], may give us an answer to the questions remained to be solved.

## 4. Materials and Methods

### 4.1. Animals and Gametes

The ascidian *Halocynthia roretzi* Type C, which had been aqua-cultured in Onagawa Bay, Japan, was purchased from a fisherman and stored in the Research Center for Marine Biology, Graduate School of Life Sciences, Tohoku University. Sperm and eggs were collected as described previously [16,27].

### 4.2. Fertilization Experiments

The effect of the ubiquitin-activating enzyme inhibitor PYR-41 (Sigma–Aldrich; ethyl 4-(4-((5-nitrofuran-2-yl)methylene)-3,5-dioxopyrazolidin-1-yl)benzoate) [36] on the fertilization of *H. roretzi* was carried out according to the method described previously [16,27]. Briefly, PYR-41 dissolved in DMSO at a concentration of 30 mM was diluted with filtered seawater buffered with 20 mM HEPES (pH 8.0). A small volume (10 µL) of sperm suspension was added to buffered artificial seawater (460 mM NaCl, 10 mM KCl, 10 mM CaCl_2_, 50 mM MgCl_2_, buffered with 50 mM Tris/HCl (pH 8.2)) containing PYR-41 and preincubated for 10 min at 13 °C. After preincubation, 20 µL of egg suspension was gently mixed and incubated for 1 h at 13 °C in a total volume of 300 µL using 48-well multiwell dishes. The fertilization ratio was determined based on the expansion of the perivitelline space at 1 h after insemination as described previously [16,27].

### 4.3. cDNA Cloning of HrUBA1 and HrUBA6

One hundred milligrams each of the testis, ovary, gill, muscles, and intestine was excised from adult *H. roretzi* type C and homogenized in 1 mL of RNAzol^®^RT (Molecular Research Center, Inc.) on ice. The homogenate was mixed with 0.4 mL of DEPC water and incubated for 15 min at room temperature, which was followed by centrifugation at 12,000× *g* for 15 min. The supernatant (1 mL) was mixed with 0.4 mL of 75% ethanol and allowed to stand for 10 min. Total RNA was precipitated by centrifugation (12,000× *g*, 8 min) and washed with 0.4 mL of 75% ethanol, followed by centrifugation (8000× *g*, 3 min). This process was repeated twice. The washed RNA was dissolved in 30 µL of DEPC water and used as a total RNA preparation.

The total RNA solution was reverse transcribed using a Super Script III First-Strand Synthesis System kit (Invitrogen) according to the manufacturer’s protocol. RT–PCR was carried out using the primers listed in Table 2 as described previously [37]. Two primers corresponding to the sequence of *EF-1α*, *UBA1* and *UBA6* genes (Table 2) were used to check the cDNA. The *HrUBA1*, *HrUBA6*, and *HrEF-1α* genes were identified as described in the text by using the ANISEED database (http://www.aniseed.cnrs.fr/aniseed/ (accessed on 15 May 2023)) [38,39]. Testis cDNAs of *HrUBA1* and *HrUBA6* were subjected to TA cloning using a pCR2-1-TOPO TA cloning kit (Invitrogen). The DNA sequences were determined by using a Big Dye Terminator v3.1 Cycle Sequence Kit (Applied Biosystems (Waltham, MA, USA)).

### 4.4. Molecular Phylogenetic Analysis

Amino acid sequences of UBA1 (E1) and UBA6 (E1L2) from humans, mice, and the ascidian *H. roretzi* were aligned by Clustal Omega, and the phylogenetic tree was made by the neighbor-joining method by MEGAX [40,41]. The accession numbers of the nucleotide sequences used are as follows: HrUBA1, LC767911 (this study); HrUBA6, LC767912 (this study); HsUBA1, NP_695012.1; MmUBA1, NP_001129557.1; HsUBA2, NP_005490.1; MmUBA2, NP_057891.1; HsUBA3, NP_003959.3; MmUBA3, NP_035796.2; HsUBA4, NP_055299.1; MmUBA4, NP_001153802.1; HsUBA5, NP_079094.1; MmUBA5, NP_079968.2; HsUBA6, NP_060697.4; MmUBA6, NP_766300.1; HsUBA7, NP_003326.2; MmUBA7, NP_076227.1.

### 4.5. Real-Time PCR

One hundred milligrams each of the testis, ovary, gill, muscles, and intestine was excised from adult *H. roretzi* type C, and their RNAs were extracted by RNAzol (Cosmo Bio). The total RNA solution was reverse transcribed using the Super Script III First-Strand Synthesis System (Invitrogen). RT–PCR was carried out as described previously [37]. For real-time PCR, mRNAs were prepared using a Dynabeads mRNA purification kit (Thermo Fisher Scientific). cDNAs were used as a template, and the following primers (Table 3) were designed based on the gene model in the ANISEED database. SYBR Premix ExTaq II ROX plus (TaKaRa) was used for analysis using a StepOnePlus (ABI). The comparative CT method was used to analyze the relative expression level of mRNA, assuming that the expression level in the testis was defined as 1.0.

### 4.6. Production and Purification of Antibodies

Two peptides corresponding to the sequences of the UBA1-specific region (CKPEMAQNGKKDNKEPDID) and UBA6-specific region (CEAKVPDFVPSNKRIETD) (see Appendix A) were covalently cross-linked to keyhole limpet hemocyanin using MBS (*m*-maleimidobenzoyl-*N*-hydroxysuccinimide ester) and immunized to rabbits. After 4 immunizations, titers were checked by ELISA (see Appendix A). Then, antisera were collected and purified by antigenic peptide-immobilized beads. Peptide synthesis, immunization, antibody purification, and ELISA were performed by Sigma–Aldrich Co. (Tokyo, Japan).

### 4.7. Immunocytochemistry

Sperm and eggs were collected from *H. roretzi* and incubated with a fixative solution (4% paraformaldehyde, 0.5 M NaCl, 200 mM MOPS) for 1 h at room temperature. The fixed sperm were diluted with PBST, placed on MAS-coated glass slides Super Frost White MAS (Matsunami), and allowed to stand for 20 min at room temperature to attach to the glass slides. The sperm supernatant and egg fixation solution were removed and washed three times with PBST for 5 min. For blocking, samples were incubated in 1% BSA in PBST for 1 h at room temperature. Primary antibodies were added at a 900-fold dilution in 1% BSA in PBST and allowed to react overnight at 4 °C. Samples were then washed three times for 15 min in PBST. Secondary antibodies Alexa Fluor 594-conjugated anti-rabbit IgG antibody (for eggs), Alexa Fluor 488-conjugated anti-rabbit IgG antibody (for sperm), and DAPI (5 mg/mL) were diluted 1000-fold in 1% BSA in PBST and added. In addition, for eggs, a 1000-fold dilution of Alexa Fluor 488 Phalloidin was added. Sperm were incubated at room temperature for 1 h, and eggs were incubated at 4 °C overnight. The sperm were then washed four times with PBST for 5 min and mounted with ProLong Gold antifade reagent (Life Technologies). The eggs were mounted with SlowFade Gold antifade reagent (Life Technologies).

## Figures and Tables

**Figure 1 ijms-24-10662-f001:**
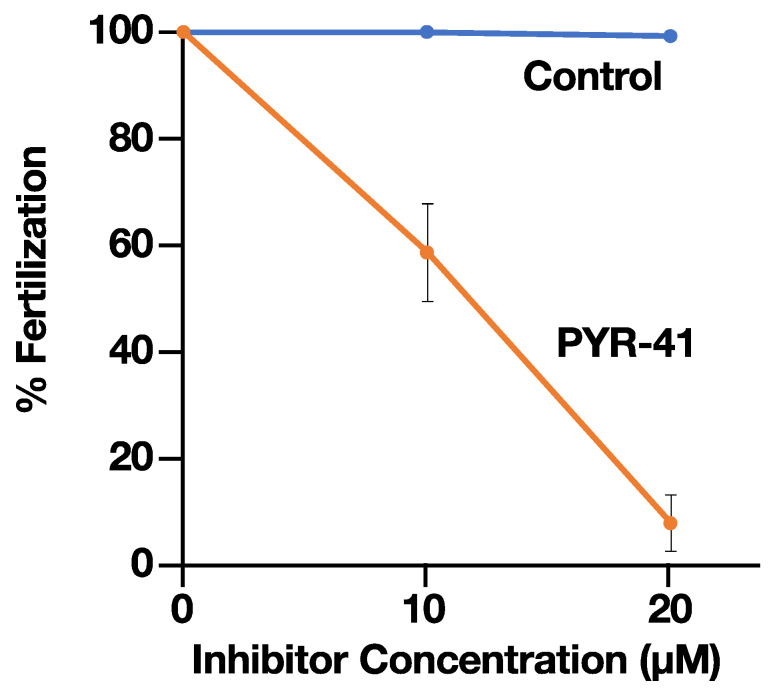
Effect of the Ub-activating enzyme inhibitor PYR-41 on fertilization of *H. roretzi*. Orange line, the effect of PYR-41; blue line, the effect of DMSO as a vehicle (Control). The 10 µM inhibitor contained 0.03% DMSO, while the 20 µM inhibitor contained 0.06% DMSO. Error bar, SD (*n* = 3). The fertilization rate was determined based on the expansion of the perivitelline space as described previously [16].

**Figure 2 ijms-24-10662-f002:**
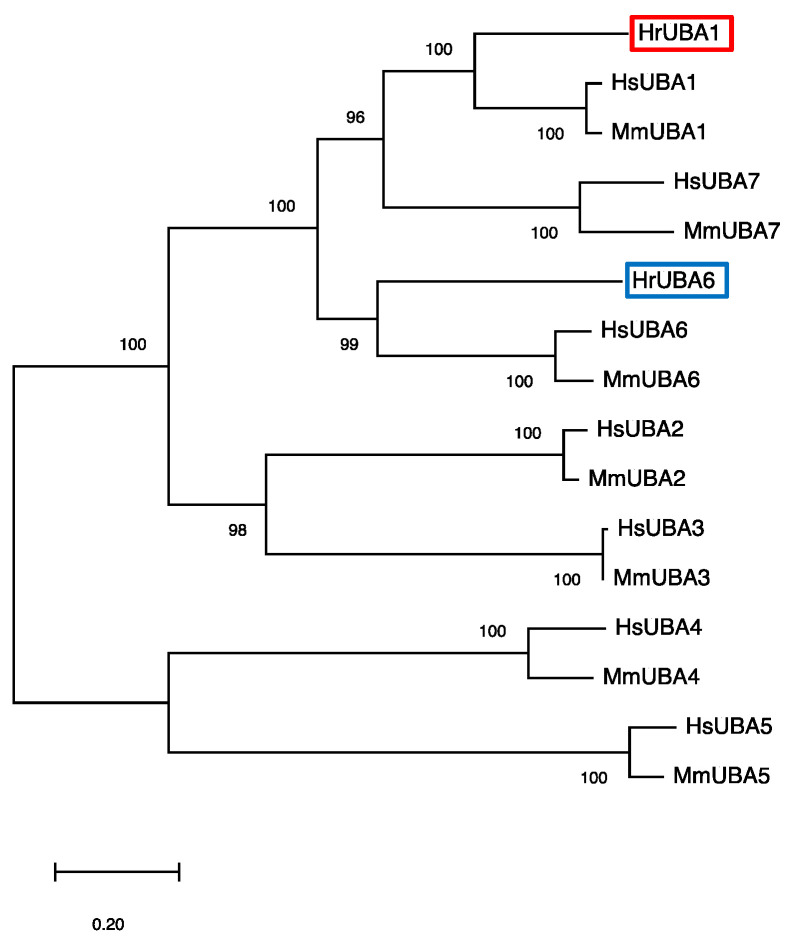
Phylogenetic tree of amino acid sequences of UBA1, UBA2, UBA3, UBA4, UBA5, UBA6, and UBA 7 from humans and mice in addition to HrUBA1 (red box) and HrUBA6 (blue box). Amino acid sequences of UBA1 (E1) and UBA6 (E1L2) from humans, mice, and the ascidian *H. roretzi* were aligned by Clustal Omega, and the phylogenetic tree was illustrated as described in the Section 4.

**Figure 3 ijms-24-10662-f003:**
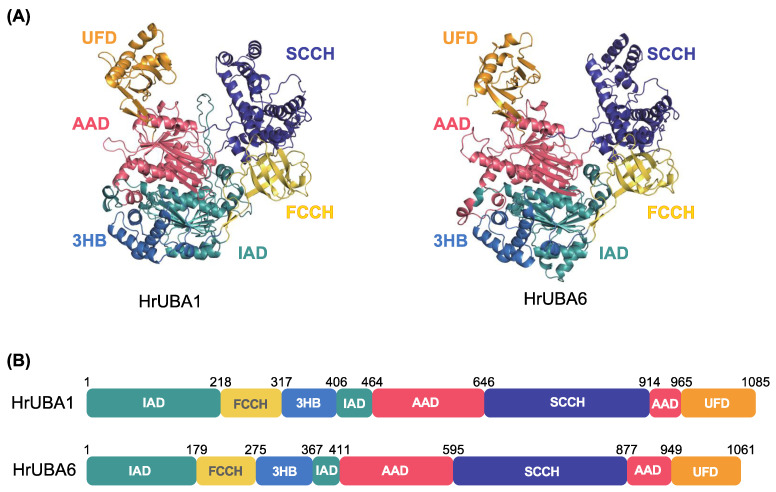
Structural visualization of HrUBA1 and HrUBA6. (**A**) AlphaFold2 structure prediction of HrUBA1 (left) and HrUBA6 (right). Models of HrUBA1 and HrUBA6 were generated from the local copy of Alphafold2 [30] and visualized using PyMOL by Schrödinger (https://pymol.org/2/ (accessed on 15 May 2023)) (**B**) HrUBA1 and HrUBA6 comprise the inactive and active adenylation domain (IAD and AAD), the first and second catalytic cysteine half domain (FCCH and SCCH), the 3-helix-bundle (3HB) and the ubiquitin fold domain (UFD) [28,29].

**Figure 4 ijms-24-10662-f004:**
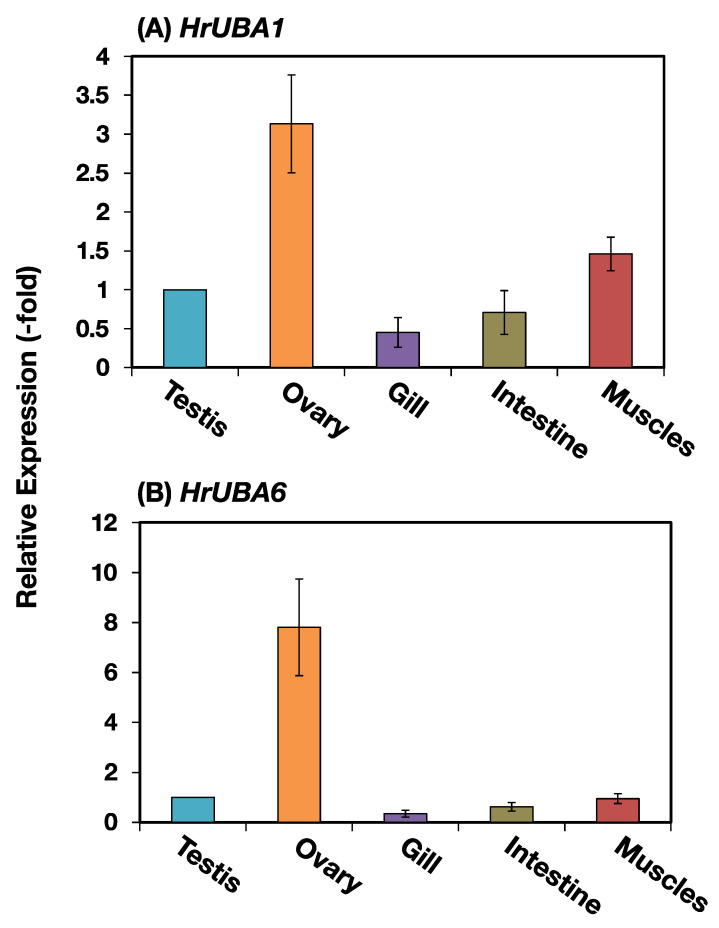
Expression of *HrUBA1* and *HrUBA6* mRNAs in respective organs by real-time PCR. cDNAs were prepared from the testis, ovary, gill, intestine, and muscles, and real-time PCR was carried out using respective cDNA preparation as a template and specific primers to *HrUBA1* (**A**) and *HrUBA6* (**B**). Transcription levels of *HrUBA1* and *HrUBA6* were normalized to the expression level of *HrEF-1α*. Respective columns indicate the relative expression, assuming the value in the testis as 1.0. Error bar, SD.

**Figure 5 ijms-24-10662-f005:**
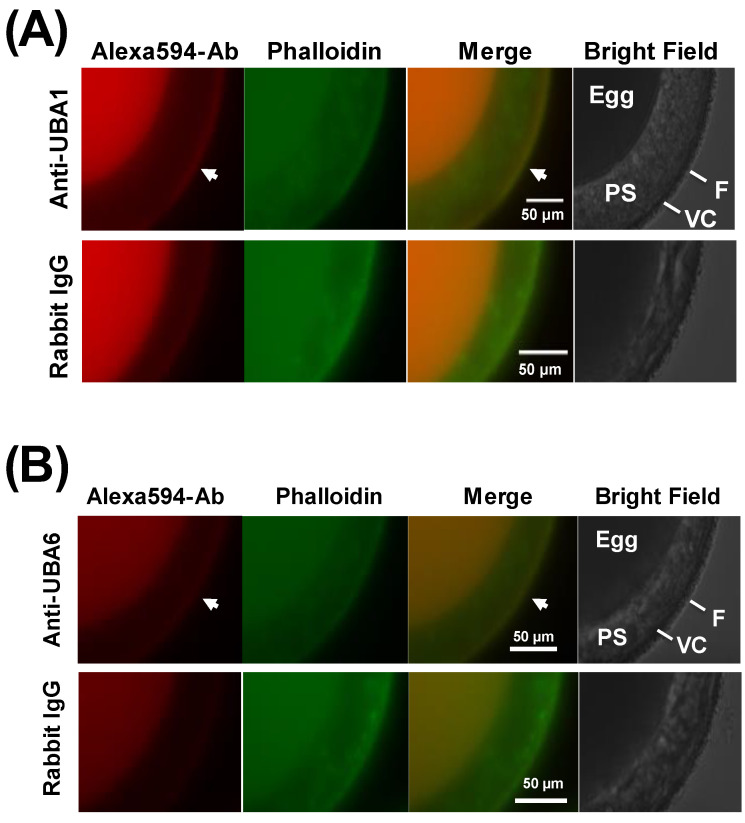
Localization of ascidian unfertilized eggs HrUBA1 and HrUBA6. Immunocytochemistry by using anti-HrUBA1 (**A**) or anti-HrUBA6 (**B**) antibody (upper panels) and control rabbit IgG (lower panels). Bars indicate 50 µm. PS, perivitelline space; F, follicle cells. Actin was stained with phalloidin. Follicle cells (and VC) of unfertilized eggs were stained (arrow).

**Figure 6 ijms-24-10662-f006:**
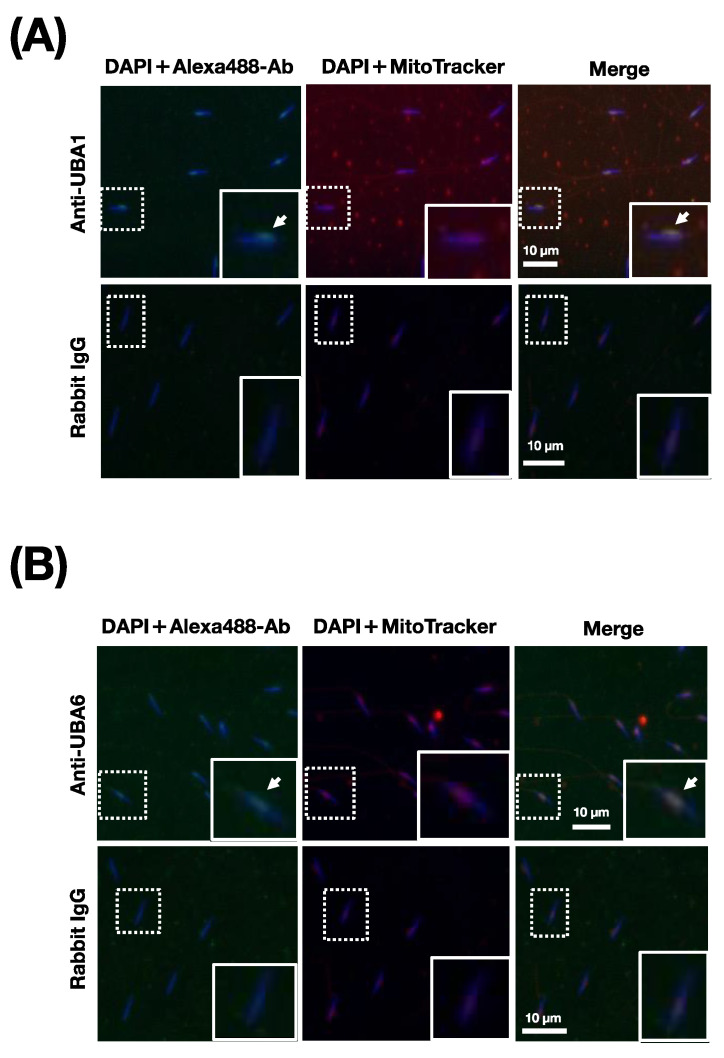
Localization of HrUBA1 and HrUBA6 in sperm. Immunocytochemistry of ascidian sperm with anti-HrUBA1 (**A**) or anti-HrUBA6 (**B**) antibody (upper panels) and rabbit IgG (lower panels). Inset is an enlarged view of the area indicated by the broken line box. Bars indicate 10 µm. DNA was stained with DAPI, and mitochondria were stained with MitoTracker. The sperm mitochondrial region was stained.

**Figure 7 ijms-24-10662-f007:**
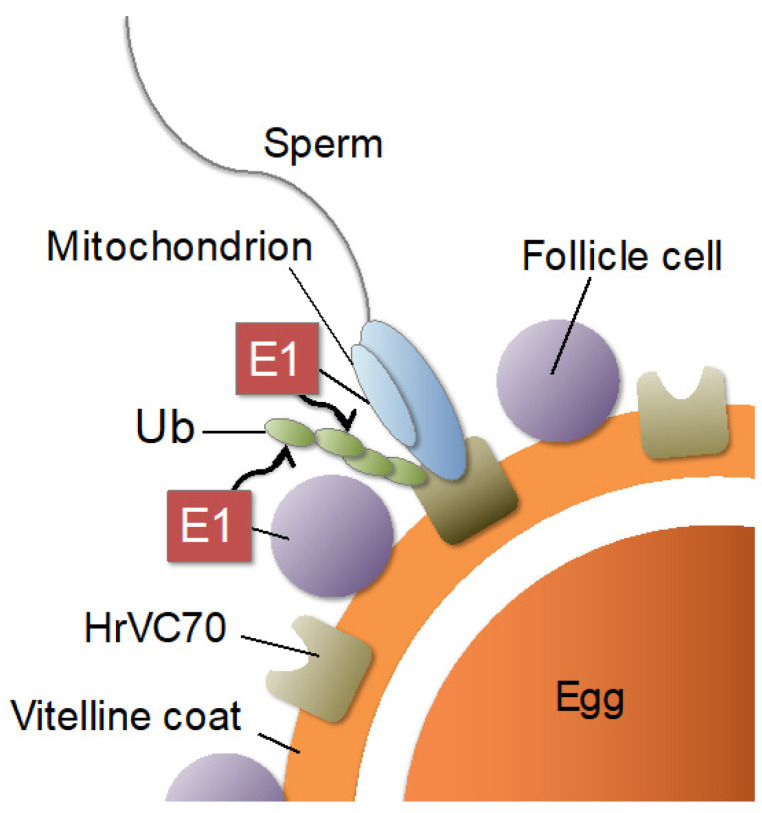
Working hypothesis for localization and role in fertilization of ubiquitin-activating enzyme E1 (UBA1/6) in the ascidian *Halocynthia roretzi*.

**Table 1 ijms-24-10662-t001:** Gene models of *UBA* homologs in *H. roretzi*.

No.	Gene Model in *H. roretzi* Genomic Database	Possible Genes	Predicted Gene by Phylogenetic Tree
1	Harore.CG.MTP2014.S314.g06093.01.p	*UBA1*, *UBA6*, *UBA7*	*UBA6* homolog
2	Harore.CG.MTP2014.S33.g15526.01.p	*MOCS3*, *UBA1*, *UBA2*	*UBA4* homolog
3	Harore.CG.MTP2014.S334.g12180.01.p	*NAE1*, *ASE1*, *UBA1*	(not assigned)
4	Harore.CG.MTP2014.S338.g06192.01.p	*UBA1*, *UBA2*, *UBA3*	*UBA2* homolog
5	Harore.CG.MTP2014.S410.g06699.01.p	*UBA1*, *UBA2*, *UBA3*	*UBA3* homolog
6	Harore.CG.MTP2014.S60.g02316.01.p	*UBA1*, *UBA6*, *UBA7*	*UBA1* homolog
7	Harore.CG.MTP2014.S91.g14409.01.p	*NAE1*, *SAE1*, *UBA1*	(not assigned)

**Table 2 ijms-24-10662-t002:** Primers used for cDNA cloning of *UBA1* and *UBA6* and for *EF-1α* genes.

Primer Name	Sequence
HrEF-1α-FWD	5′-GGGAAGAGTGGAGACTGGA-3′
HrEF-1α-REV	5′-CTTACCAGAGCGACGATCG-3′
HrUBA1-FWD (#10)	5′-ACCGATGAGTAAACTAATTTGACG-3′
HrUBA1-REV (#11)	5′-TTCTACGTCTTCACCGTCTTGG-3′
HrUBA1-FWD (#15)	5′-AGACAGAAAAAGTTTACAATGACG-3′
HrUBA1-REV (#13)	5′-ATGAATTTGGAACGTTGTCATGG-3′
HrUBA6-FWD(#1)	5′-AAAGAAGTATACAAGAGGGTAG-3′
HrUBA6-REV (#8)	5′-AGTGAA ATTAGAAAGA ACAGTG-3′

**Table 3 ijms-24-10662-t003:** Specific primers used for real-time PCR of the *UBA1*, *UBA6*, and *EF-1α* genes.

Primer Name	Sequence
RT-UBA1-FWD	5′-AATGTCGCCATGCTGTACTC-3′
RT-UBA1-REV	5′-CAAATCTCGAACACGAGAGC-3′
RT-HrUBA6-FWD	5′-CATGCAGGGTACAAGAATGG-3′
RT-UBA6-REV	5′-TGACCGAGTCTTTTCGAATG-3′
RT-EF-1α -FWD	5′-ATCCCGGAAACATTTCAAAG-3′
RT-EF-1α-REV	5′-ACTCTTTGGGTTTTCCTCCA-3′

## Data Availability

Nucleotide sequences reported here have been submitted to DDBJ. The accession number of HrUBA1 is LC67911 and that of HrUBA6 is LC767912.

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
