# Peer review of "Involvement in Fertilization and Expression of Gamete Ubiquitin-Activating Enzymes UBA1 and UBA6 in the Ascidian *Halocynthia roretzi"

_ijms, 2023, doi:10.3390/ijms241310662_

Round 1
Reviewer 1 Report
Introduction
- The data of HrUBA1 & 6 on fertilization process
Results
1. Figure 1 : should have the symbols of blue and red in the figure
2. Figure 2 : check the correctio of red box
3. Figure 4 : Change to the color chart
1. The writing pattern in manuscripts should be adjusted to a passive voice style, it will be better.
2. The pattern of abbreviation such as Line 51-52, please check with the format of journal
3. About the abbreviation, such as Line 148-152 , should write the full-name first?
Author Response
<Reviewer 1>
(Comment 1) Figure 1: should have the symbols of blue and red in the figure.
(Response) Figure 1 was changed to line graph using color symbols as suggested.
(Comment 2) Figure 2: check the correct in (position?) of red box.
(Response) Red boxes were moved to the correct position. In order to distinguish HrUBA1 and HrUBA6, HrUBA1 was indicated by red box and HrUBA6 was indicated by blue box.
(Comment 3) Figure 4: Change to the color chart.
(Response) In Figure 4, the monochromic chart was changed to the color chart.
(Comment 4) The writing pattern in manuscripts should be adjusted to a passive voice style, it will be better.
(Response) Most active voice sentences such as “we …” were modified to passive voice as suggested.
(Comment 5) The pattern of abbreviation such as Line 51-52, please check with the format of journal.
(Response) FAT10 is a diubiquitin-like modifier previously named by others. So, we adopted this abbreviation. As described in the text “human leukocyte antigen (HLA)-F-adjacent transcript 10 (FAT10), …”, three letters used for abbreviation were indicated by underline.
(Comment 6) About the abbreviation, such as Line 148-152, should write the full-name first?
(Response) Abbreviations were spelled out and indicated before respective abbreviations.
Reviewer 2 Report
In the article titled “Gamete Ubiquitin-Activating Enzymes UBA1 and UBA6 Involved in Fertilization of the Ascidian Halocynthia roretzi” the authors propose that HrUBA1, HrUBA6, or both in the sperm head mitochondrial region and follicle cells may be involved in the ubiquitination of HrVC70, which is responsible for the fertilization of H. roretzi.
I can reconsider the possibility of publication after a major revision.
My suggestions are the following:
The title is not clear
The abstract needs revision in terms of clarity.
All work needs revision of English
The sequence of information in the introduction should be written more clearly and connecting the sentences with a logical thread so that the reader can better understand the topic
In Figure 1, make the % fertilization homogeneous with the wording in the caption: The fertilization ratio was determined based on the expansion of the perivitelline space as described previously [16].
In the discussion define the limitations of the study and future prospects
Extensive editing of English language required
Author Response
<Reviewer2>
(Comment 1) The title is not clear.
(Response) The title was modified as follows “Involvement in Fertilization and Expression of Gamete Ubiquitin-Activating Enzymes UBA1 and UBA6 in the Ascidian Halocynthia roretzi” in order to clarify the contents.
(Comment 2) The abstract needs revision in terms of clarity.
(Response) Most active voice sentences were modified to passive voice sentence except “Here, we show…” and the contents were slightly modified. In order to help readers’ understanding, we added some explanation to “vitelline coat” and “ascidian”, such as “vitelline coat, a proteinaceous egg coat,” and “ascidian (Urochordata)”, since ascidians are taxonomically classified into Urochordata. In addition, the expression “de novo” was added to Line 21, to clarify whether E1 activity is really necessary during fertilization or not. The original abstract has been checked and modified by all the authors and English proofreading company (American Journal Expert). So, we wouldn’t like to change the expression. We will reconsider further modifications, if the reviewer specifically points out the sentences or points, which are hard to understand. Please let us know. But, we hope that our current expression and style would be understandable for the reproductive biologists.
(Comment 3) All work needs revision of English.
(Response) English of this manuscript has been edited by AJE (American Journal Expert). So, we believe that English itself is grammatically correct.
(Comments 4) The sequence of information in the introduction should be written more clearly and connecting the sentences with a logical thread so that the reader can better understand the topic.
(Response) We slightly modified for better understanding. But, we do not like to drastically change the contents. We think that the entire story of this paper is reasonable.
(Comment 5) In Figure 1, make the % fertilization homogeneous with the wording in the caption: The fertilization ratio was determined based on the expansion of the perivitelline space as described previously [16].
(Response) The ordinate was changed to “% Fertilization”, and the expression was changed from columns to line graph. In this species, the expansion of perivitelline space in a fertilized egg is always observed after gamete fusion (fertilization), which was written in our previous paper [16]. So, this is a good criterion for fertilization.
(Comment 6) In the discussion define the limitations of the study and future prospects.
(Response) We added a limitation of our future experiment such as genome editing at the end of discussion. “Although genome editing experiments of E1, E2, and E3 would be difficult due to inability of culturing juvenile to a sexually matured adult in aquarium, several specific inhibitors against the UPS [33], including deubiquitinating enzyme [34, 35], may give us an answer to the questions remained to be solved.”
(Comment 7) Extensive editing of English language required.
(Response) English of this manuscript has been edited by AJE (American Journal Expert). So, we believe that English itself is grammatically correct. We submit a certificate by AJE as a reference, where one or more English proofreading editors modified our English. The expression “en masse”, which was recommended by AJE, was deleted because it is not usual to use this expression in our scientific field.
Reviewer 3 Report
Hitoshi Sawada and co-authors present a quality and well-written experimental manuscript focused on gamete ubiquitin-activating enzymes UBA1 and UBA6 involved in fertilization of the Ascidian Halocynthia roretzi.
Authors report here that show that PYR-41, a UBA inhibitor, strongly inhibited the fertilization of H. roretzi. To identify that the cDNA cloning of UBA1 and UBA6 from H. roretzi gonads was carried out, and their 3D protein structures were predicted to be very similar to those of human UBA1 and UBA6, respectively, based on AlphaFold2. These two genes were transcribed in the ovary and testis and other organs, among which the expression of both was highest in the ovary. Immunocytochemistry showed that these enzymes are localized on the sperm head around a mitochondrial region and the follicle cells surrounding the VC. These results led them to propose that HrUBA1, HrUBA6, or both in the sperm head mitochondrial region and follicle cells may be involved in the ubiquitination of HrVC70, which is responsible for the fertilization of H. roretzi.
To address whether E1 activity plays a key role in the process of ascidian fertilization, authors first investigated the effect of an E1 inhibitor on fertilization. Since the participation of E1 in the fertilization of H. roretzi was suggested, cDNAs of E1 enzymes expressed in the testis were cloned, and the domain architecture and 3D structures were predicted. Localization of E1s was also investigated by immunocytochemistry using affinity-purified antibodies.
Authors studied: effect of E1 inhibitor on fertilization; identification of the UBA1 and UBA6 genes in H. roretzi; transcriptional levels of HrUBA1 and HrUBA6 in several organs; localization of HrUBA1 and HrUBA6 in sperm and eggs;
Finally, authors conclude that they demonstrated that the mRNAs of two ubiquitin-activating enzymes, HrUBA1 and HrUBA6, are expressed in the ovary and testis in the ascidian Halocynthia roretzi. Immunocytochemical studies demonstrated that HrUBA1/6 appears to be localized around a sperm mitochondrion, which localizes on the lateral side of the sperm head. It has been reported that the ascidian sperm surface over a mitochondrion binds to the surface of the VC, which allows sperm penetration through the VC using the motive force of mitochondrial sliding through the flagellum. Therefore, localization of the ubiquitin-proteasome system in the mitochondrial region, which is involved in making a small hole for sperm passage of the VC, seems reasonable.
==============================
Overall, the manuscript is highly valuable for the scientific community and should be accepted for publication after the corrections are made.
Other comments:
1) Please check for typos throughout the manuscript.
2) Authors are kindly encouraged to cite the following article that describes the function and therapeutic targeting of the ubiquitin-specific protease which can also be involved in fertilization process.
DOI: 10.3390/cancers14225539
Author Response
<Reviewer 3>
(Comment 1) Please check for typos throughout the manuscript.
(Response) Thank you for kind advice. We checked.
(Comment 2) Authors are kindly encouraged to cite the following article that describes the function and therapeutic targeting of the ubiquitin-specific protease which can also be involved in fertilization process. DOI: 10.3390/cancers14225539
(Response) Although USP is not directly related in this paper, we added the citation of this paper at the end of Discussion as follows. In connection with this, three references [33-35] were inserted.
“Although genome editing experiments of E1, E2, and E3 would be difficult due to inability of culturing juvenile to a sexually matured adult in aquarium, several specific inhibitors against the UPS [33], including deubiquitinating enzyme [34, 35], may give us an answer to the questions remained to be solved.”
Round 2
Reviewer 2 Report
Accepted in the present form
Minor editing of English language required